# Wild Boar (*Sus scrofa*)—*Fascioloides magna* Interaction from the Perspective of the MHC Genes

**DOI:** 10.3390/pathogens11111359

**Published:** 2022-11-16

**Authors:** Dean Konjević, Vlatka Erman, Miljenko Bujanić, Ida Svetličić, Haidi Arbanasić, Snježana Lubura Strunjak, Ana Galov

**Affiliations:** 1Veterinary Faculty, University of Zagreb, Heinzelova 55, 10000 Zagreb, Croatia; 2State Inspectorate, Šubićeva 29, 10000 Zagreb, Croatia; 3Department of Biology, Faculty of Science, University of Zagreb, Rooseveltov trg 6, 10000 Zagreb, Croatia; 4Department of Mathematics, Faculty of Science, University of Zagreb, Bijenička cesta 30, 10000 Zagreb, Croatia

**Keywords:** fascioloidosis, swine leucocyte antigens, host–parasite interactions

## Abstract

Fascioloidosis is a parasitic disease caused by a trematode *Fascioloides magna*. Since major histocompatibility complex (MHC) genes play an important role in the immune response, the aim of this study was to compare the potential differences in MHC class II SLA-DRB1 exon 2 genes between wild boar populations from infected (cases) and non-infected areas (controls). During the winter of 2021, a total of 136 wild boar tissue samples were collected, 39 cases and 97 controls. DNA was extracted and sequenced using the Illumina platform. Differences in distributions of allele combinations were calculated using the Chi-Square test for homogeneity and between proportions using the large-sample test and Fisher–Irwin test. Analysis revealed 19 previously described swine leucocyte antigen (SLA) alleles. The number of polymorphic sites was 79 (29.6%), with 99 mutations in total. Nucleotide diversity π was estimated at 0.11. Proportions of the alleles SLA-DRB1*12:05 (*p* = 0.0008379) and SLA-DRB1*0101 (*p* = 0.0002825) were statistically significantly higher in controls, and proportions of the SLA-DRB1*0602 (*p* = 0.006059) and SLA-DRB1*0901 (*p* = 0.0006601) in cases. Alleles SLA-DRB1*04:09, SLA-DRB1*0501, SLA-DRB1*11:09, and SLA-DRB1*1301 were detected only in cases, while SLA-DRB1*0404, SLA-DRB1*0701, SLA-DRB1*02:10, and SLA-DRB1*04:08 were present only in controls. We did not confirm the existence of specific alleles that could be linked to *F. magna* infection. Detected high variability of the MHC class II SLA-DRB1 exon 2 genes indicate high resistance potential against various pathogens.

## 1. Introduction

Non-native trematode *Fascioloides magna* has begun its invasion of Europe after the import of infected white-tailed (*Odocoileus virginianus*) and wapiti deer (*Cervus elaphus canadensis*) from North America to Italy [1]. However, according to recent studies, fascioloidosis did not spread further from Italy but was rather introduced at least two more times to the areas of the Czech Republic and Danube floodplain forests [1,2,3]. Among final hosts, three different types have emerged in Europe so far, definitive (red deer, *Cervus elaphus*, and fallow deer, *Dama dama*), aberrant (roe deer, *Capreolus capreolus*, and mouflon, *Ovis musimon*), and dead-end (wild boar, *Sus scrofa*) [1]. Contact with each of them resulted in different clinical pictures, gross lesions, and outcomes of infection, but also marked the development of new host–parasite interactions. Descriptions of fascioloidosis in wild boar as well as in feral pigs in the USA are rare [1,4,5,6,7,8,9]. In general, when metacercarias are ingested by wild suids, juvenile *F. magna* can reach the liver, but rarely matures and usually provokes the formation of a thick-walled pseudocyst that will eventually kill the fluke. The prevalence in suids in infected areas can be low [1] but can also reach up to 69% [10]. Up to now, none of the studies has described excretion of eggs by infected wild boars. Moreover, despite the fact that various trematodes can infect humans [11], to the authors’ knowledge, there are no reports on human fascioloidosis.

Major Histocompatibility Complex (MHC) genes are a highly polymorphic part of the vertebrate genome, coding for molecules that present pathogen-derived antigens to T-cells. Given their role in the adaptive immune response to parasites, MHC genes are under the direct influence of parasite-mediated selection [12]. Therefore, many studies have tried to link the MHC characteristics with susceptibility/resistance to disease in wild animals [12,13,14]. So far, two main hypotheses related to their role in fighting various pathogens have been developed. First one states that higher variability of MHC genes leads to increased resistance to pathogens [15], and the other one claims that the existence of specific (also called rare) alleles is responsible for increased susceptibility/resistance to disease [16]. The significance of MHC variation in adaptation to parasites has been validated in a number of studies, e.g., [17,18,19,20,21]. In pigs, MHC complex, also called the swine leucocyte antigen (SLA), is mapped on seven chromosomes and was studied extensively [22]. Highly polymorphic DRB locus (part of the SLA class II) is comprised of one functional gene (DRB1) and four pseudogenes (DRB2, DRB3, DRB4, and DRB5) [23]. Even though the majority of MHC studies in pigs were conducted on domestic breeds, several of them analyzed MHC variability in wild boars [24,25,26]. Due to many difficulties, like unknown genetic background or health status (infections with other parasites) and the constant impact of various environmental factors, studies on the relation between MHC genes and disease in wildlife are still facing many constraints. Since fascioloidosis is a disease caused by a parasite non-native to Europe, it represents a good opportunity to mimic experimental conditions in a free-living population. In other words, animals living in non-infected areas can be effectively used as negative controls since, most probably, neither have they nor their ancestors ever been in contact with *F. magna*. Regarding wild boar genetics, it is also necessary to mention that previous studies on wild boar subspecies claimed the existence of up to 16 subspecies in Europe based on morphological characteristics [27]. More recent studies, using both morphological and microsatellite analysis, have reported the existence of one subspecies in the area of Croatia [28,29].

The aim of this study was to compare the variability and potential existence of specific alleles of the MHC-DRB locus exon 2 between wild boars that originate from *F. magna* infected and non-infected areas.

## 2. Results 

### 2.1. Parasitological Findings

The results of the parasitological analysis of livers are presented in Table 1. Pseudocysts, fluke migratory channels, and adult/juvenile flukes were found in four wild boar livers. The remaining positive livers showed only irregular liver surface, loss of translucency of Glisson’s capsule, fibrin deposits, and traces of black pigment (iron porphyrin) (Figure 1).

### 2.2. MHC Genes 

The results of the analysis of DRB exon 2 sequences in 136 wild boars are presented in Table 2. Analysis revealed 19 previously described swine leucocyte antigen (SLA) alleles. The number of polymorphic sites across all detected sequences was 79 (29.6%), with 99 mutations in total. Nucleotide diversity π was estimated at 0.11, while the average number of nucleotide differences k was estimated at 29.23. The detected alleles could be translated into unique amino acid sequences with 38 (42.7%) polymorphic sites. The overall nucleotide evolutionary distance and the amino acid evolutionary distance were 0.18 ± 0.03 and 0.35 ± 0.08, respectively. Among the analyzed individuals, 26 were homozygotes (10 in the positive group and 16 in the negative). Twenty-two individuals showed locus duplication, present exclusively in the allelic combination of SLA-DRB1*04:10 and SLA-DRB1*06:07. Twenty of them were heterozygotes (two from cases and 18 from the control group) while the remaining two animals (from the cases group) had locus duplication in homozygotic form, hence, they had four alleles in total. 

### 2.3. Comparison between Cases and Controls

The test statistic value was 79.561, with *p* = 0.003746, leading to the rejection of the null hypothesis. Proportions of the alleles SLA-DRB1*12:05 (*p* = 0.0008379) and SLA-DRB1*0101 (*p* = 0.0002825) were statistically significantly higher in wild boars from the control group. On the other hand, the proportions of the alleles SLA-DRB1*0602 (*p* = 0.006059) and SLA-DRB1*0901 (*p* = 0.0006601) were statistically significantly higher in wild boars from the cases group. Interestingly, eight alleles were detected only in one group of wild boars, four in cases and four in controls. Alleles SLA-DRB1*04:09, SLA-DRB1*0501, SLA-DRB1*11:09, and SLA-DRB1*1301 were detected only in cases group, while alleles SLA-DRB1*0404, SLA-DRB1*0701, SLA-DRB1*02:10, and SLA-DRB1*04:08 were present only in the control group (Figure 2). We found no statistically significant difference when comparing different genotypes.

## 3. Discussion

An increasing wild boar population in Europe is of extreme importance from many points, including damage to agriculture and forests, conflicts with humans (collisions with vehicles, colonization of urban areas), and potential role in disease transmission [30,31,32,33,34]. 

In this study, among 136 wild boars, we have identified relatively high MHC variability, with 19 different SLA DRB alleles. Similarly, a study on four different wild boar populations from Italy (two populations), Hungary, and Poland has revealed 18 different alleles in 57 animals [24]. A previous study in Croatia detected 20 alleles in 133 analyzed animals, describing eight new alleles that were not previously reported in GenBank [26]. Previously acknowledged locus duplication [26] was also detected in 22 individuals analyzed in this study, all of which had allelic combinations of SLA-DRB1*04:10 and SLA-DRB1*06:07 linked as a two-locus haplotype. As MHC genes play an important role in the immune system, it is believed that high polymorphism is maintained by pathogen-driven selection [15]. This is also supported by the fact that wild boars are potential carriers of various pathogens, which will lead to more polymorphic MHC genes. Such higher variability (heterozygosity) should result in hosts’ ability to detect a wider range of antigens. However, despite the role of high heterozygosity, there are descriptions of direct selection on the MHC genes when combating new or coevolving pathogens, indicating the importance of renewal of genetic variations resulting from mutation, immigration of new genes, or recombination [35,36]. In our study, polymorphic sites across all detected sequences were 79 (29.6%), with 99 mutations in total. The formation of new, rare alleles can offer a temporary selective advantage over the common ones [16]. Since the continental wild boar population in Croatia is thought to share common pathogens, it was expected that the presence of *Fascioloides magna* could result in differences in MHC alleles between the cases and the control group. In our study, we found statistically higher proportions of two haplotypes in control wild boars compared to cases and vice versa. Besides that, we also found four alleles that occur only in the cases group and four alleles present only in the control group. Unfortunately, unlike in red deer, where it was possible to link the intensity of infection with different alleles, and where temporary named allele DRB_ref06 was associated only with low-intensity infected animals [37], in wild boars, due to the specific nature of host–parasite interaction, the majority of findings were only represented by fibrin deposits and traces of black, iron porphyrin pigment, which did not enable us to quantify the intensity of infection. Additionally, differently from definitive and aberrant hosts, wild boar as dead-end hosts already show resistance to *F. magna* by encapsulating it in a thick-walled pseudocyst. Both, along with known difficulties to find MHC-parasite associations [38], made it much more difficult to evaluate the potential role of specific alleles, thus, limiting the potential of this study. 

To conclude, in our study, we did not confirm the presence of specific alleles that could be linked to susceptibility/resistance to *F. magna*. The high variability of MHC genes in wild boars offers potentially high resistance against various pathogens. If any of alleles that are detected only in the cases group is related to interactions with *F. magna,* it will increase in frequency in the future [39], as a consequence of dynamic host–parasite coevolution [16,40]. If the current protective role of wild boars in the case of fascioloidosis [41] changes in the future, due to their vast distribution area and large populations, wild boars will become an important factor in the epidemiology of fascioloidosis. Therefore, it is very important to monitor the development of this host–parasite interaction. This is the first study that compared MHC gene characteristics with *F. magna* infection.

## 4. Material and Methods

### 4.1. Locations, Parasitological Analysis and Sampling

A case-control design using a large wild model (wild boar) was applied. Animals originating from the areas where *F. magna* was not yet detected (Medvednica Nature Park and hunting ground “Črnovšćak”) were considered negative (control group). Contrary, animals from infected areas (hunting grounds “Opeke II”, “Podunavlje-Podravlje”, and “Breznica”) were considered positive (cases group). Even if gross lesions or flukes were not detected in livers, due to the rapid turnover of generations in wild boar and possible infection of their parents, combined with the long-lasting presence of *F. magna* in red deer populations in the same areas, all animals from infected areas were considered as cases. Since all animals were collected by the kindness of hunters following regular game management operations, we used non-probability convenient sampling. The study was approved by the Committee on the Ethics of the University of Zagreb, Faculty of Veterinary Medicine (Class: 640-01/18-17/60; No. 251-61-44-18-02).

Relief of the study areas was characterized as lowland (Črnovšćak, Opeke II, and Podunavlje-Podravlje) and hilly (Medvednica Nature Park and Breznica). Forest associations in lowland habitats were typical for humid areas and included *Salicetum albo*—*triandrae*, *Salicetum albae*, *Salici albae-Populetum nigrae,* and *Fraxino-Ulnetum laevis* associations presenting better conditions for fluke developmental cycle, while on the hilly habitats, dominant associations were *Carpiniom betuli Illyricum*, *Quercion robori Petraea*, *Carici sylvaticae*—*Quercetum petraea*, *Querco petraea*—*Carpinetum illyricum*, *Querco petraea*—*Carpinetum illyricum* var. *Fagus silvatica*, indicating less humid areas. Dominant large game species are red deer (except in non-infected areas), wild boar, roe deer, red fox, and golden jackal. According to some studies, the prevalence of *F. magna*-infected red deer varies between 60 to approx. 80% [42,43,44]. 

In total, we have collected samples from 136 wild boars (39 cases; 97 controls). Each liver was analyzed macroscopically for potential signs of infection: Enlargement of the liver, opaque Glisson’s capsule, fibrin deposits, traces of black pigment (iron-porphyrin), and irregular liver surface. Following external inspection, livers were cut to approx. 2 cm-thick slices and thoroughly examined from both sides for the presence of fluke’s migratory channels, pseudocysts, and juvenile/adult flukes. 

### 4.2. Molecular Analysis

Genomic DNA was extracted using Wizard Genomic DNA Purification Kit (Promega), following the manufacturer’s protocol. Sequencing for typing on the Illumina platform was performed in the Novogene facility (UK) as follows. The 267 bp long coding sequence of DRB exon II was amplified using the degenerate primer pair DRB1F-22 and DRB1R+284 [45] connected with sample-specific barcodes. PCR amplicons were size-selected on 2% agarose gel, pooled, and purified. The purified products were end-repaired, A-tailed, and ligated with Illumina sequencing adapters. Constructed DNA libraries were quantified with Qubit and real-time PCR, after which they were inspected with a bioanalyzer for size distribution and sequenced on the Illumina MySeq platform to generate 250 bp long paired-end reads. The resulting paired-end reads were merged using the software FLASH [46]. Low-quality sequences were filtered out using the Qiime quality control script [47], and UCHIME [48] algorithm was utilized for the chimera removal. DRB genotyping was performed using the AmpliSAS web tool [49], which considers only the first 5000 sequence reads to reduce the computational load. Default AmpliSAS parameters were selected for Illumina sequencing technology: 1% substitution error rate and 0.001% indel error rate. The minimum dominant clustering threshold was 25%, and the minimum per amplicon frequency filtering threshold was set to 3%. Non-coding sequences and frameshifts were automatically discarded. Detected sequences were inspected and aligned in BioEdit software [50]. The number of polymorphic sites and the average number of nucleotide differences were calculated using DNASp [51]. Nucleotide and amino acid evolutionary distances were estimated in MEGA X according to the substitution model with the lowest BIC score [52]. Sequences were compared with previously known SLA alleles using the BLAST tool (https://blast.ncbi.nlm.nih.gov, accessed on 14 September 2022). 

### 4.3. Statistical Analysis

All data were analyzed using software R (R × 64 3.6.1). The Chi-Square test for homogeneity was used to test the differences in distributions of allele combinations between wild boars from infected and *F. magna*-free areas. Since alleles SLA-DRB1*06:07 and SLA-DRB1*04:10 appeared only in combination as a two-locus haplotype, they were analyzed as one. Consequently, a total of 18 alleles were present in the two sample groups. The large-sample test based on the normal approximation to the binomial and Fisher–Irwin test were used to test differences between proportions of particular alleles (or particular combinations of alleles). 

## Figures and Tables

**Figure 1 pathogens-11-01359-f001:**
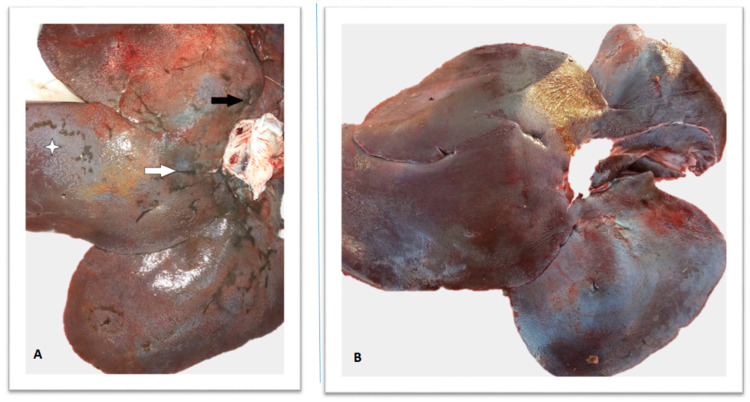
(**A**) positive wild boar liver. White arrow indicates iron-porphyrin traces, four-point star indicates loss of translucency, while black arrow indicates surface irregularities. (**B**) Negative wild boar liver.

**Figure 2 pathogens-11-01359-f002:**
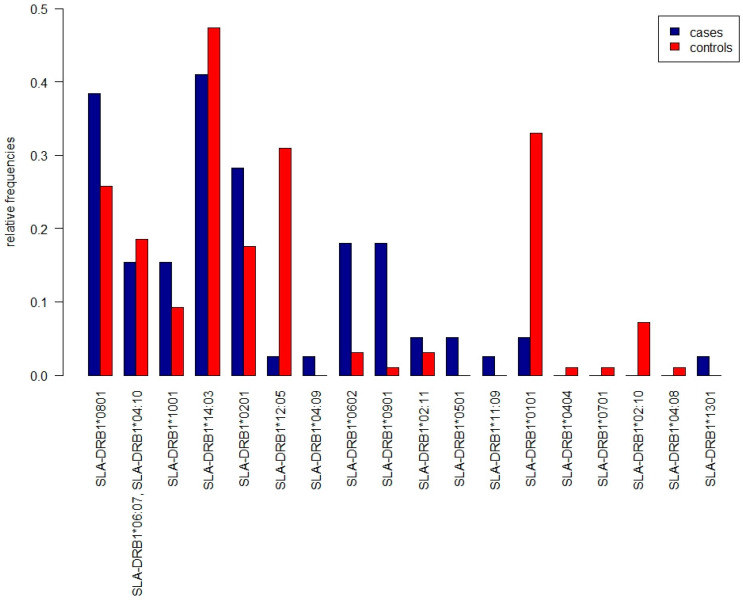
Relative frequencies of the MHC-DRB alleles, blue columns represent cases, red columns represent controls. Note the frequencies of the alleles SLA-DRB1*04:09, SLA-DRB1*0501, SLA-DRB1*11:09, and SLA-DRB1*1301 appear only in the cases group.

**Table 1 pathogens-11-01359-t001:** Results of parasitological analysis of liver. Prevalence in the cases group was 48.72%.

	Infected Area (Cases)	Non-infected Area (Controls)
Total sample	39	97
Positive	19	0
Negative	20	97

**Table 2 pathogens-11-01359-t002:** Frequencies and relative frequencies of the MHC class II SLA-DRB1 exon 2 alleles found in the case and control group.

	Cases (N = 39)		Controls (N = 97)	
Alleles	Frequencies	Relative Frequencies	Frequencies	Relative Frequencies
SLA-DRB1*0801	15	0.384615	25	0.257732
SLA-DRB1*06:07 SLA-DRB1*04:10	6	0.153846	18	0.185567
SLA-DRB1*1001	6	0.153846	9	0.092784
SLA-DRB1*14:03	16	0.410256	46	0.474227
SLA-DRB1*0201	11	0.282051	17	0.175258
SLA-DRB1*12:05	1	0.025641	30	0.309278
SLA-DRB1*04:09	1	0.025641	0	0
SLA-DRB1*0602	7	0.179487	3	0.030928
SLA-DRB1*0901	7	0.179487	1	0.010309
SLA-DRB1*02:11	2	0.051282	3	0.030928
SLA-DRB1*0501	2	0.051282	0	0
SLA-DRB1*11:09	1	0.025641	0	0
SLA-DRB1*0101	2	0.051282	32	0.329897
SLA-DRB1*0404	0	0	1	0.010309
SLA-DRB1*0701	0	0	1	0.010309
SLA-DRB1*02:10	0	0	7	0.072165
SLA-DRB1*04:08	0	0	1	0.010309
SLA-DRB1*1301	1	0.025641	0	0

Alleles SLA-DRB1*06:07 and SLA-DRB1*04:10 appear only in combination and are analyzed as one.

## Data Availability

Not applicable.

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
