# Peer review of "Wild Boar (Sus scrofa)—Fascioloides magna Interaction from the Perspective of the MHC Genes"

_pathogens, 2022, doi:10.3390/pathogens11111359_

Round 1
Reviewer 1 Report
The introductory part of the abstract is large. In the abstract you should explore the results more. The conclusion of the abstract should be the same as the discussion. Some of this literature should be removed.
All key words must be different from the title.
line 36, avoid personal or third party stuff. remove it.
line 57, should in the introduction explore this application further and remove much of the story quoted above.
All table and figure titles should be self explanatory and more complete.
In the discussion review only should be avoided. In line 110, that paragraph is a revision. It should have the authors discuss their results and compare with other results. Make the discussion again.
The discussion is short. There is hardly any discussion of the results.
In materials and methods the writing should be impersonal.
On line 182, why this difference in cases and control? There is a big difference. This may affect the result.
Reviewer 2 Report
Abstract
Line 18.. please add the name of genes analyzed.
Alaso mention the total no of SNPs observed/analyzed.
line 19-20. please add the number of case and controls?
when this study was conducted and in which specific area?
If possible, provide the genotype/allelfrequency in parenthesis for case and control for better understanding.
Results
Figure 1. Is not it better that you show the liver of case and control to compare the morphological and histological changes, if any.
Figure 2. Can standard error bars be added? Was there some sttaistical analysis perfomed on this data? If yes, please mention it in caption or foot note.
Discussion
How these results would be interesting for the scientific community? Please discuss.
What is the limitation of this study?
Please briefly discuss the previous studies on this topic and compare your results with them.
What is the conclusion of this study? Please add.
Introduction
Has the genotype at MHC genes of wild boar populations has been reported previously? If yes, briefly discuss it here.
What is the potenial impact of Fascioloides magna infection in boar to human? Any zoonotic aspect?
Round 2
Reviewer 1 Report
The article has improved and deserves to be published.